# eQTL mapping using allele-specific count data is computationally feasible, powerful, and provides individual-specific estimates of genetic effects

Vasyl Zhabotynsky[1]*, Licai Huang[2], Paul Little[3], Yi-Juan Hu[4], Fernando Pardo-Manuel de Villena[5,6], Fei Zou[1,5], Wei Sun[1,3,7]*

**1** Department of Biostatistics, University of North Carolina, Chapel Hill, North Carolina, United States of America, **2** Quantitative Sciences Program, The University of Texas MD Anderson Cancer Center and UTHealth Graduate School of Biomedical Sciences, Houston, Texas, United States of America, **3** Public Health Science Division, Fred Hutchinson Cancer Research Center, Seattle, Washington, United States of America, **4** Department of Biostatistics and Bioinformatics, Emory University, Atlanta, Georgia, United States of America, **5** Department of Genetics, University of North Carolina, Chapel Hill, North Carolina, United States of America, **6** Lineberger Comprehensive Cancer Center, University of North Carolina, Chapel Hill, North Carolina, United States of America, **7** Department of Biostatistics, University of Washington, Seattle, Washington, United States of America

* vasyl@unc.edu (VZ); wsun@fredhutch.org (WS)

**Data Availability Statement:** All data underlying the findings are fully available without restriction. The RNA-seq data generated by the Geuvadis

## Abstract

Using information from allele-specific gene expression (ASE) can improve the power to map gene expression quantitative trait loci (eQTLs). However, such practice has been limited, partly due to computational challenges and lack of clarification on the size of power gain or new findings besides improved power. We have developed geoP, a computationally efficient method to estimate permutation p-values, which makes it computationally feasible to perform eQTL mapping with ASE counts for large cohorts. We have applied geoP to map eQTLs in 28 human tissues using the data from the Genotype-Tissue Expression (GTEx) project. We demonstrate that using ASE data not only substantially improve the power to detect eQTLs, but also allow us to quantify individual-specific genetic effects, which can be used to study the variation of eQTL effect sizes with respect to other covariates. We also compared two popular methods for eQTL mapping with ASE: TReCASE and RASQUAL. TReCASE is ten times or more faster than RASQUAL and it provides more robust type I error control.

## Author summary

An effective approach to study the genetic basis of complex diseases is to assess the associations between genetic variants and gene expression. The statistical power to detect such associations can be improved by using gene expression measurement for each parental allele, though with the price of higher computational cost. We have developed a new method to improve the computational efficiency to make it computationally feasible to

consortium are available at http://www.ebi.ac.uk/arrayexpress/experiments/E-GEUV-1/samples/. The RNA-seq data from GTEx are available at https://gtexportal.org/home/datasets. All the data underlying Figs 1-5 are provided in S6 Table. Additional intermediate data and pipeline can be found at https://github.com/Sun-lab/asSeq_pipelines.

**Funding:** VZ, LH, PL, YJH, FZ, and WS were supported in part by NIGMS (https://www.nigms.nih.gov/) grant R01 GM105785. VZ, FZ, and F.P.-M.d.V were supported in part by NIEHS grant P42ES031007. VZ was also supported in part by NIEHS (https://www.niehs.nih.gov/) 5T32ES007018. The funders had no role in study design, data collection and analysis, decision to publish, or preparation of the manuscript.

**Competing interests:** The authors have declared that no competing interests exist.

conduct such analyses in large cohorts. We applied our method to analyze the genetic and gene expression data from 28 human tissues and reported a comprehensive resource on the genetic basis of gene expression. We also demonstrated an advantage to use gene expression of individual alleles: quantification of the genetic effect on gene expression for each individual. Such individual-specific estimates of genetic effects allowed us to explore the dynamics of genetic effects, e.g., variation of genetic effect with respect to age. Finally, we also evaluated the underlying model assumption of different methods and pointed out the model assumption adopted by a popular method could lead to more false discoveries than expected.

## Introduction

Mapping gene expression quantitative trait loci (eQTLs) is an effective and popular approach to study the function of genetic variants [1]. An eQTL study may assess the associations between the expression of tens of thousands of genes and the genotypes of millions of single nucleotide variants (SNPs). This daunting computational task can be accomplished efficiently by some elegant computational methods, such as MatrixEQTL [2] or FastQTL [3]. The core of such methods is a linear regression model for each (gene, SNP) pair, where the response variable is gene expression (after appropriate transformation if needed) and the covariates include SNP genotype together with possible confounders such as batch effects. These linear regression methods use the total expression of each gene across all the alleles (e.g., summation of gene expression from maternal and paternal allele for a diploid genome). RNA-seq data can also measure allele-specific gene expression (ASE). Exploiting ASE information can substantially improve the power of eQTL mapping [4]. More precisely, ASE can inform the mapping for a *cis*-acting eQTL that affects gene expression in an allele-specific manner (e.g., a genetic variant on the maternal allele only influences the gene expression of the maternal allele) [5]. Most eQTLs detectable with a sample size of a few hundred are local eQTLs around the gene of interest (e.g., within 500kb of the gene), and the vast majority of the local eQTLs are *cis*-acting eQTLs [4, 5].

A few computational methods have been developed for eQTL mapping using both total expression and ASE, including TReCASE (Total Read Count + ASE) [4], CHT (combined haplotype test) [6], and RASQUAL (Robust Allele Specific Quantitation and Quality Control) [7]. TReCASE [4] was the first method of this kind. It was later extended to account for the uncertainty to phase the eQTL SNP and the exonic SNPs in the gene body [8]. CHT allows extra over-dispersion in total expression and accounts for genotyping errors. RASQUAL implemented some elegant strategies to account for sequencing/mapping errors, reference bias, genotyping errors, as well as phasing errors. It has been demonstrated that CHT has similar performance as RASQUAL but is computationally more demanding [7], and thus we will not consider CHT in this work.

The application of eQTL mapping using ASE is hindered by two computational challenges. One is the computational cost of appropriate multiple testing correction for local eQTL mapping. Most of the local SNPs of a gene have highly correlated genotypes due to linkage disequilibrium. Therefore, the effective number of independent tests is much smaller than the number of local SNPs. A naive multiple testing correction method assumes the number of tests is the number of local SNPs and thus is too conservative. Calculating permutation p-values is an effective solution to account for linkage disequilibrium of local SNPs [3]. However, it is computationally prohibitive to run TReCASE or RASQUAL for thousands of permutations

per (gene, SNP) pair. To address this challenge, we have developed a computational method to approximate permutation p-values by estimating the effective number of independent tests, which varies with respect to p-value cutoffs. We name this method as "geoP" based on a geometric interpretation of permutation p-values [9]. Another computational challenge is the preparation of ASE, which requires access to raw data (e.g., bam files). Since raw data are often too large to be stored in a local computing environment, it is desirable to use raw data saved on cloud. To this end, we have developed a workflow to extract all the inputs for TReCASE from raw data saved locally or on cloud.

Equipped by our geoP method for permutation p-value estimation and our cloud-based data processing pipeline, we performed eQTL mapping in 28 tissues from Genotype-Tissue Expression (GTEx) study [1]. Our results substantially expand the eQTL findings. Using a permutation p-value cutoff of 0.01 (corresponding to FDR around 1%), we detected 20–100% more eGenes (genes with at least one significant eQTL) than the most recent GTEx study [1], where ASE was not used in eQTL mapping. We have also made thorough comparisons of TReCASE versus RASQUAL. TReCASE controls type I error well while RASQUAL may lose type I error control, especially for the genes with multiple heterozygous exonic SNPs. We also provide explanation by examining the likelihood function of RASQUAL. Furthermore, RASQUAL requires 10–100 times of computational time of TReCASE, making it computationally very challenging for large scale eQTL studies. Overall, our work delivers a resource of eQTL findings in 28 GTEx tissues and provides computational tools and guidance for future eQTL studies.

## Results

### eQTL mapping using TReCASE

The inputs to our workflow of data preparation include raw data of gene expression (i.e., bam files of RNA-seq data), gene annotation (i.e., the beginnings and ends of each exon of each gene), and a list of phased heterozygous SNPs for each individual. Such phasing information can be obtained by computationally phasing unphased genotype data [10], which is usually accurate enough since we only use the phase information with a relatively short distance (e.g., 500kb). Our workflow, a docker image that can be used either locally or in a cloud setting, extracts total read count (or total fragment count for paired-end reads) and ASE using these inputs (Fig 1a). If an RNA-seq read overlaps with more than one heterozygous SNPs, it will be counted multiple times if ASE is quantified per SNP. Therefore, it is more accurate to measure ASE per haplotype rather than per SNP (Fig 1b). For organisms with more diverse parental genomes (e.g., F1 mice), more sophisticated methods are needed to accurately align the RNA-seq reads to each haplotype [11].

To calculate permutation p-values without brute force permutations, we estimate the effective number of independent tests, denoted by $M_{\text{eff}}$, and then calculate the permutation p-value corresponding to a nominal p-value $p$ by $\max(pM_{\text{eff}}, 1)$. Several methods have been proposed to estimate $M_{\text{eff}}$. For example, eigenMT [12] estimates $M_{\text{eff}}$ as the minimum number of sample eigenvalues required to explain a proportion of the sample variance. This estimate is constant and does not change with respect to the nominal p-value cutoff. Based on a geometric interpretation of permutation test, we have shown conceptually and empirically that $M_{\text{eff}}$ increases as the nominal p-value cutoff decreases [9]. In fact, permutation p-value estimates based on eigenMT tend to be conservative around permutation p-value cutoff 0.01 and is more accurate for more stringent cutoffs such as 0.001 (Fig 1c and 1d). Because eQTL signals are abundant genome-wide, a permutation p-value cutoff of 0.01 often corresponds to false

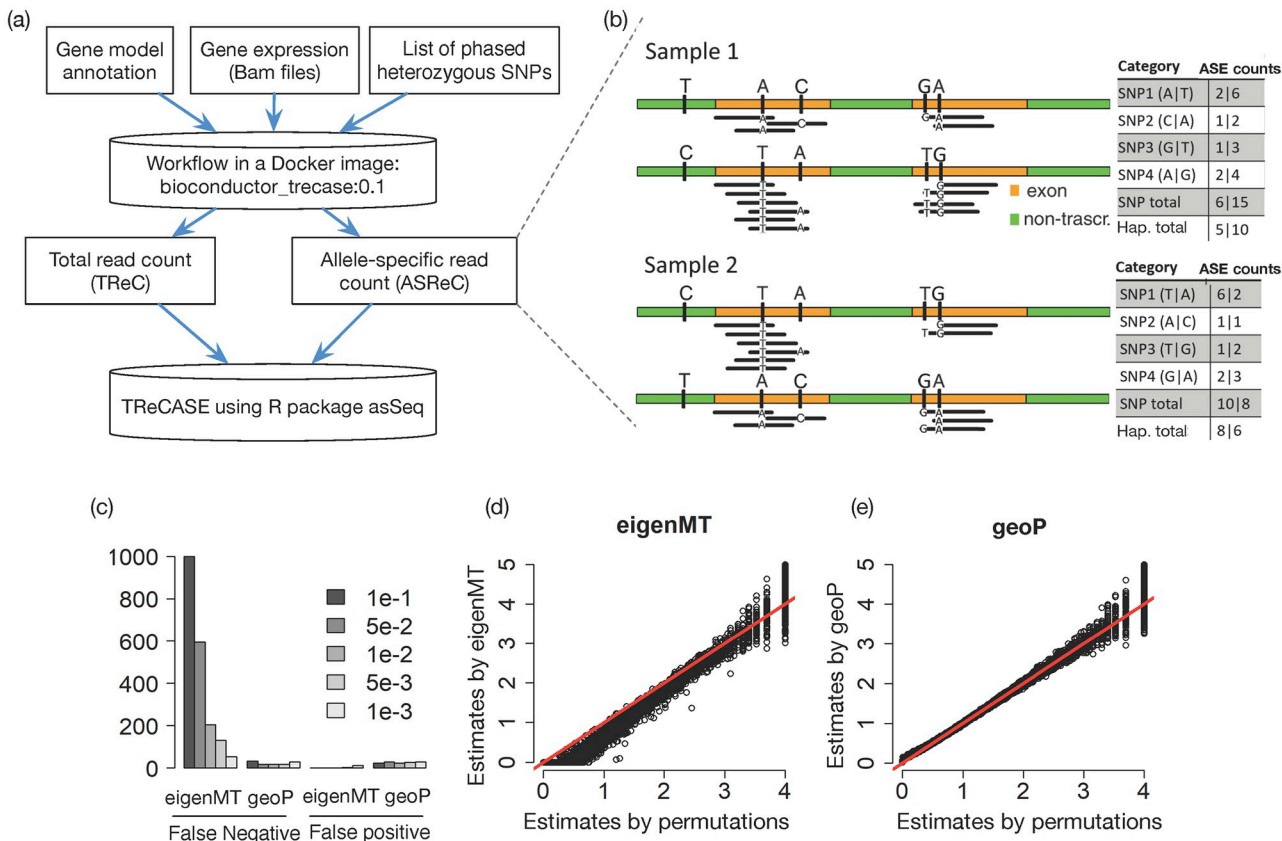

**Fig 1. Overview of our pipeline and geoP method.** (a) A workflow starting with raw data on the cloud to extract gene expression information, followed by eQTL mapping using TReCASE. (b) Quantification of ASE by counting allele-specific reads. The table on the right side shows the count for each SNP and the summation (SNP total) or the total count on haplotype level (ASE count) and the latter avoids double counting. (c-e) Comparison of permutation p-values estimated by eigenMT or geoP, versus "true" values generated by 10,000 permutations, using the eQTL data of 14,566 genes from the GEUVADIS dataset [13]. (c) The number of false negatives or false positives at each permutation p-value cutoff labeled in the legend. A gene is considered as false negative (positive) at a cutoff $\alpha$ if its permutation p-value estimate is larger (smaller) than $\alpha$, while the "true" value from 10,000 permutations is equal to or smaller (larger) than $\alpha$. (d-e) A scatter plot of -$\log_{10}$(permutation p-value) estimated by 10,000 permutations (x-axis) versus the estimates by eigenMT or geoP.

discovery rate around 1%, and thus the accuracy of permutation p-value estimates around 0.01 is important.

We propose a method called geoP to estimate $M_{\texttt{eff}}$ as a function of nominal p-value cutoff. For each gene, we fit a linear model of its (transformed) expression versus SNP genotype of the most significant local eQTL as well as other covariates. Next, we generate $k$ parametric bootstrap samples ($k = 100$ by default) based on this linear model while plugging in different eQTL effect sizes. For each bootstrap sample, we calculate the minimum p-value across all the local SNPs, as well as the corresponding permutation p-value using up to 1,000 permutations. Then we fit a logistic regression with sample size $k$ to predict permutation p-values using log transformed minimum nominal p-value. At first sight, this is counter-intuitive because geoP does not avoid permutations; instead, it uses more permutations than directly estimating permutation p-values. This is computationally sensible because geoP uses computationally much more efficient linear regression instead of TReCASE. In fact, the time needed to calculate permutation p-values by geoP is less than running TReCASE itself (Table A6 in S1 Text). With extra

computations as a price, geoP provides more accurate estimates of permutation p-values than eigenMT (Fig 1c–1e).

## TReCASE identifies 20–100% or more eGenes than linear model across 28 tissues of the GTEx study

We reanalyzed the GTEx v8 data in 28 tissues (with sample size from 175 to 706) to identify local eQTLs using three methods: linear model by MatrixEQTL, TReC that only use total read counts, and TReCASE. For each gene, the mapping window is defined as the gene body plus 500kb window flanking the gene body on either side. After calculating permutation p-values using geoP for each gene, multiple testing across genes can be corrected by choosing a permutation p-value cutoff to control q values [14]. Since there are strong eQTL signals for most of the genes, q-values, which take into account of the proportion of eGenes, are often smaller than permutation p-values. For example, a q-value cutoff 0.05 may correspond to a permutation p-value cutoff larger than 0.1. To stay on the conservative side, we used permutation p-value 0.01 as cutoff in our analysis and the corresponding q-values are around 0.01 as well.

We first compare the number of eGenes identified by MatrixEQTL versus the eGenes reported by the most recent GTEx publication [1] where the same linear model as the one implemented in MatrixEQTL was used. The GTEx analysis [1] is slightly different from ours in two aspects. It uses a mapping window of 1Mb around the transcription starting site and the permutation p-values are estimated by up to 10,000 permutations. In contrast, our mapping window is gene body plus 500kb flanking regions and we estimate permutation p-values using geoP. Despite these minor differences, the number of eGenes reported by the two pipelines (when we use MatrixEQTL for eQTL mapping) are highly consistent (Fig 2a). The total number of eGenes identified by MatrixEQTL ranges 90% to 100% (median 98%) of the number of eGenes identified by GTEx. The percentage of overlaps among all the GTEx eGenes ranges from 76% to 90%, with median of 86%. The overlap is not extremely high because GTEx and us search different genomic regions for eQTLs, which not only affect the candidate set of eQTLs but also the number of tests, hence the calculation of permutation p-values. The additional eGenes identified by TReCASE is derived from two sources. First, without using ASE, just applying the TReC method that models read counts using a negative binomial distribution (or a Poisson distribution when appropriate) identifies more eGenes (Fig 2b). Second, adding the ASE information further increase the number of eGenes (Fig 2c).

The additional eGenes that are identified by TReCASE decreases as sample size increases. When sample size is small (around 200), the number of eGenes identified by TReCASE is almost twice of the number of eGenes identified by MatrixEQTL (Fig 2d). Among the eGenes detected by either TReCASE or MatrixEQTL, the proportions of eGenes uniquely identified by MatrixEQTL are almost 0 (Fig 2e), and thus TReCASE recovers almost all of the MatrixEQTL findings and identifies additional ones. We have also performed a down-sampling analysis using the Geuvadis dataset [13] to demonstrate that sample size matters for the benefit of using ASE in eQTL mapping. At sample size 35, MatrixEQTL cannot identify any eGene. In contrast, TReC and TReCASE identify 224 and 454 eGenes, respectively (Fig 2f). At sample size 70, TReC can double the findings of MatrixEQTL, while TReCASE can quadruple the number of findings (Fig 2f).

Additional findings from real data may not indicate power gain, but due to larger number of false discoveries. We have conducted simulation studies with different effect sizes and sample sizes to demonstrate for typical eQTL effect sizes observed in GTEx data, TReCASE can indeed reach more than 100% power gain than MatrixEQTL (Section C.2 of S1 Text).

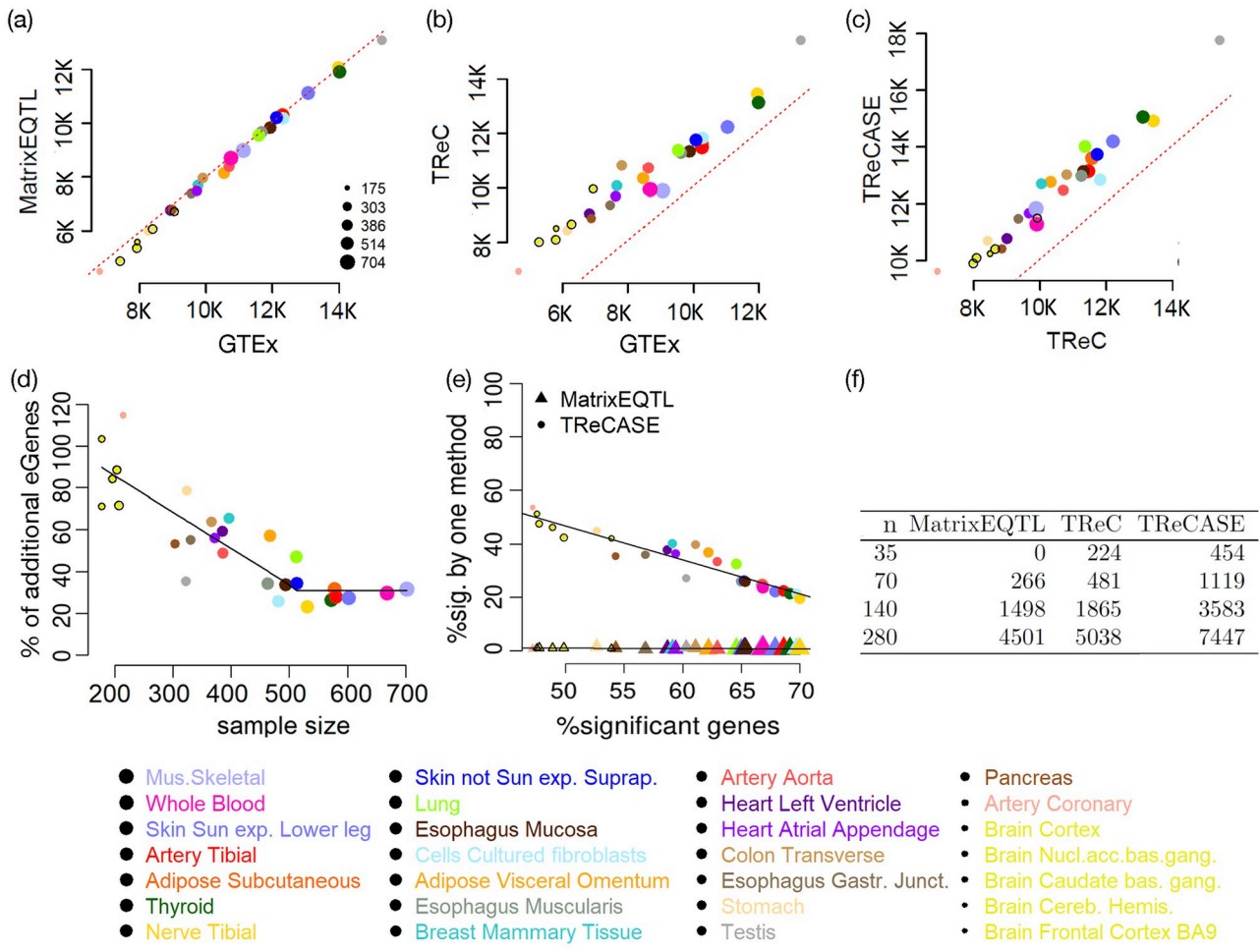

**Fig 2. Compare the number of eGenes identified by different methods using the GTEx data [1] or the Geuvadis data [13].** (a)-(c) Comparison of the number of eGenes (at permutation p-value 0.01) identified by MatrixEQTL, TReC, and TReCASE as well as reported by GTEx publication [1]. Each point represents a tissue of GTEx study. The size of a point is proportional to the sample size of the corresponding tissue. Extra black circle is added to a few smallest points to enhance their visibility. The red dotted line is a reference line of y = x. (d) The percentage of additional eGenes identified by TReCASE vs. MatrixEQTL. A piece-wise linear model fit is added to show the trend. (e) Among all the eGenes identified by either TReCASE or MatrixEQTL, the percentage reported by only one method. Two fitted line were added to show the trend. (f) The number of eGenes identified from the Geuvadis dataset [13], with sub-sampling to study the power at different sample sizes.

We also compared the number of gene-SNP pairs identified by MatrixEQTL and TReCASE and their intersections. At permutation p-value cutoff 0.01, the vast majority (96%-98%) of the gene-SNP pairs identified by MatrixEQTL can be identified by TReCASE, and number of additional gene-SNP pairs identified by TReCASE ranges from 38% to 100% of the number of gene-SNP pairs identified by MatrixEQTL (Figs A8 and A9 in S1 Text). The results are similar across several permutation cutoffs (Fig A10 in S1 Text). A few examples where the eQTL signals were identified by TReCASE but missed by MatrixEQTL were shown in Section C.3.2 of S1 Text.

## TReCASE eQTLs have similar enrichment on functional categories and GWAS hits as linear model eQTLs

The proportions of additional eGenes identified by TReCASE across the 28 tissues are consistent with what we found by simulation studies. Though it is still a fair question whether some

of the additional eQTLs identified by TReCASE are false positives. While it is beyond the scope of this paper to validate all the eQTL findings, we conducted some indirect evaluations by asking whether the eQTLs identified by TReCASE have similar enrichment on functional loci or genomic loci identified by Genome-Wide Association Studies.

We applied torus [15] to study the enrichment of eQTLs in different functional categories that were compiled by the GTEx investigators [1]. The overall enrichment patterns are consistent across the eQTLs identified by MatrixEQTL, TReC, or TReCASE when combining the results of 28 tissues (Fig 3a) or considering each tissue separately (S1 and S2 Tables). Next, for each eGene (at permutation p-value 0.01) we selected its top eQTL (the one with smallest p-value) and assessed their functional enrichment. These eQTLs were divided into a few groups based on their statistical significance by different methods. Since the number of eQTLs in each group could be too small to run torus, we quantified the enrichment by the log odds ratio for significant eQTLs in a functional category versus all the SNPs in this category. For those eGenes identified by both MatrixEQTL and TReCASE, we divided the corresponding top eQTLs into three groups: those reported by both methods, and those identified by one but not the other method (Fig 3b). We also examined the top eQTLs for the eGenes identified by one but not the other method (Fig 3c). Overall, the enrichment patterns for TReCASE and MatrixEQTL findings are very similar, though the MatrixEQTL findings tend to have higher enrichment for two categories: splice acceptor and slice donor, suggesting that TReCASE has lower power to detect isoform eQTLs than MatrixEQTL, although more specialized method should be applied to identify isoform eQTLs as done in the GTEx study [1].

We also noted that with a larger sample size, a higher fraction of eQTLs falls into one of the functional categories. After fitting a 4-parameter dose-response model of the probability that an eQTL falls into one of these categories versus sample size, we conclude that about 80% of eQTLs fall into one of the defined categories when sample size is large enough (Fig 3d). Note that these functional categories cover 56.7% of the SNPs used in eQTL mapping, which translates to an overall 1.4-fold of enrichment of eQTLs in the union of these categories.

Since the enhancer regions often vary across tissues, we expanded our study using the tissue-specific enhancer regions from EnhancerAtlas 2.0 [16], which covers five of the 28 GTEx tissues. Among three of these five tissues, the eQTL enrichment in tissue-specific enhancers is much stronger than that in the more generic definition of enhancer regions used in GTEx study. The degree of enrichment is similar for the findings of TReCASE and MatrixEQTL. Therefore, the functional enrichment results suggest that most of additional eQTL findings by TReCASE have similar functional category enrichment as those found by both methods or only by MatrixEQTL.

We also evaluated the overlap between GWAS hits and all the eQTLs identified by linear model (MatrixEQTL) or TReCASE at permutation p-value cutoff 0.01. We downloaded GWAS hits from GWAS catalog (https://www.ebi.ac.uk/gwas/docs/file-downloads, version 1.0, accessed on 11/05/2021), and considered the enrichment for all GWAS hits or for one of 21 categories (Fig A15 in S1 Text). Overall the enrichment patterns are similar across 28 tissues and between linear model (MatrixEQTL) and TReCASE. We also assessed the significance of enrichment by jacknife confidence interval. We do observer cases where GWAS hits of some categories are significantly enriched among the TReCASE eQTLs but not MatrixEQTL eQTLs. In these of such cases, the connection between GWAS categories and eQTL tissues are apparent. For example, the GWAS hits in the categories of "colon" and "mouth teeth" are enriched among the eQTLs from Colon Transverse. In some other cases, our results may indicate some unexpected connections between tissues. For example, the GWAS hits in the categories of "liver" are enriched among the eQTLs from some brain tissues.

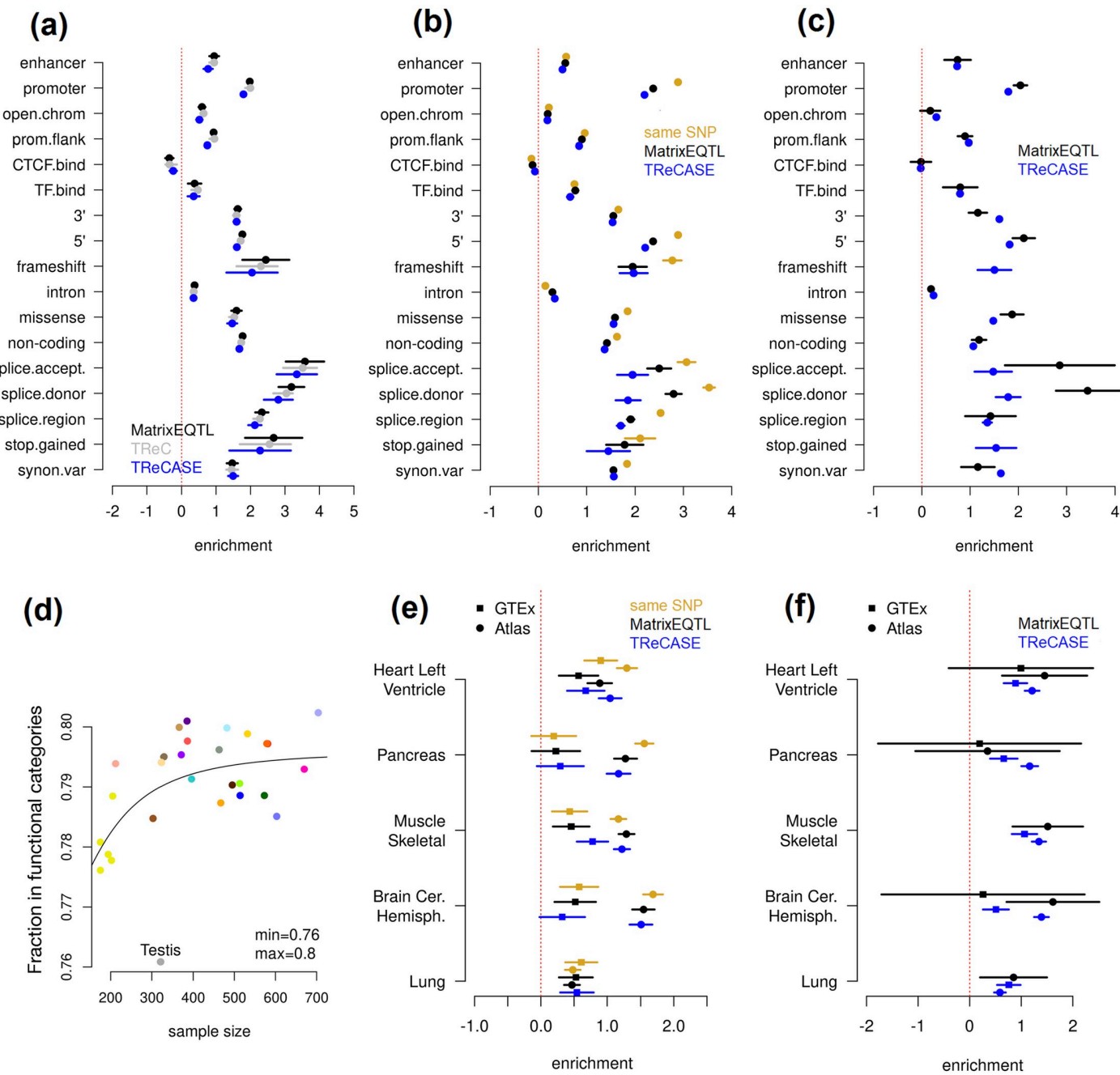

**Fig 3. Enrichment of eQTLs in functional categories using the eQTL results from 28 GTEx tissues.** In panel (a)-(c) and (e)-(f), a dot indicates point estimate, and a line indicates 95% confidence interval. (a) Enrichment evaluated using all the SNPs by torus [17] based on the eQTL results from MatrixEQTL, TReC or TReCASE. (b) Enrichment of the top eQTL per gene for the eGenes identified by both MatrixEQTL and TReCASE (permutation p-value < 0.01). The top eQTLs of these eGenes are divided into three groups, the ones reported by both methods or by one of the two methods. (c) Enrichment of the top eQTL per gene for the eGenes reported by either MatrixEQTL or TReCASE, but not both. (d) The percentage of significant eQTLs (top eQTL per eGene with permutation p-value < 0.01) in at least one functional category versus sample size in all 28 GTEx tissues. Each point is a tissue and the color coding is shown at the bottom of Fig 2. Panels (e) and (f) are analogous to panels (b) and (c), but concentrating only on enhancers in five tissues and comparing generic enhancers used in the GTEx study versus tissue-specific enhancers from EnhancerAtlas [16].

## Exploring dynamic eQTLs using individual-specific genetic effects estimated by ASE

An interesting topic in eQTL mapping is dynamic eQTLs [18], for which the genetic effect on gene expression varies with respect to another variable. These dynamic eQTLs are also referred to as context-dependent eQTLs [19] or interactions between genetic variation and environment [20]. For an eGene, we can quantify the ASE associated with each allele of the eQTL among those individuals who have heterozygous genotypes at the eQTL. The effect sizes of the eQTLs for each individual can be quantified by the proportion of gene expression from one allele (defined based on eQTL genotype), which we arbitrarily refer to as haplotype 1. We model the allele-specific read count (ASReC) from haplotype 1 by a beta-binomial distribution and associate the proportion of gene expression from haplotype 1 with covariates of interest (See Online Methods for more details). This is different from the EAGLE method [20] that uses ASE to study dynamic eQTL. EAGLE models the absolute deviation from allelic balance and thus does not need to distinguish the two haplotypes. It is more flexible since it can be applied to unphased data, though it does not fully utilize the information on the direction of the dynamic eQTLs.

An in-depth study of dynamic eQTLs warrants separate works tailored to the contents of interest. Here we mainly want to use some simple examples to illustrate that ASE has the power to deliver individual-specific eQTL effect estimates, which are very useful source to study dynamic eQTLs. We reason that when there are dynamic eQTLs, we should also see eQTL signals without conditioning on particular content. In fact, this is not a stringent requirement given that around 50–70% of all genes tested are identified as eGenes by TReCASE across the 28 GTEx tissues in our study. For each eGene, we only studied the dynamic eQTL potential for the SNP with strongest marginal eQTL signal. We explored dynamic eQTLs with respect to age or the expression of two transcription factors (TFs) *CTCF* and *TP53* since TF expression may modulate the strength of eQTLs located in TF binding sites. *CTCF* and *TP53* represent two types of TFs. *CTCF* acts as an insulator of chromatin regions and thus its function is more general and unspecific. In contrast, *TP53* has more specific (although still broad) function to respond to cellular stresses.

First, for each eGene and each conditioning variable, we fit a short model that only includes the conditioning variable, and we detected a large number of dynamic eQTLs in many tissues (Fig 4a–4c and S3 and S5 Tables). Most such dynamic eQTLs become insignificant in a long model that includes top 5 PEER (Probabilistic Estimation of Expression Residuals) factors [21] and top 2 genotype PCs (principal components) (Fig 4a–4c) that are provided by the GTEx study [1]. These results imply that the PEER factors or genotype PCs capture some latent factors that are associated with both the variable of interest and eQTL effect sizes. A potential candidate of such latent factors is cell type proportions [19, 20]. For example, for GTEx whole blood data, the proportion of neutrophil is strongly associated with the first PEER factor (Fig 4d) and age (Fig 4e). Therefore, before including the PEER factors in the model, most of the dynamic eQTLs with respect to age are likely neutrophil-specific eQTLs and their eQTL effects are associated with age because neutrophil proportion is associated with age. It is not clear what are the latent factors for the dynamic eQTLs with respect to *CTCF* or *TP53*, though the expression of both *CTCF* and *TP53* are strongly associated with the PEER factors and genotypes PCs included in the long model (Fig A16 in S1 Text).

Dynamic eQTLs can also be identified using total expression instead of ASE, for example, by adding an interaction term (e.g., an interaction between age and genetic effect) in the eQTL mapping model [19]. The advantage to use ASE is that individual-specific eQTL effects can be estimated and visualized and thus allows a more flexible model on the relation between eQTL

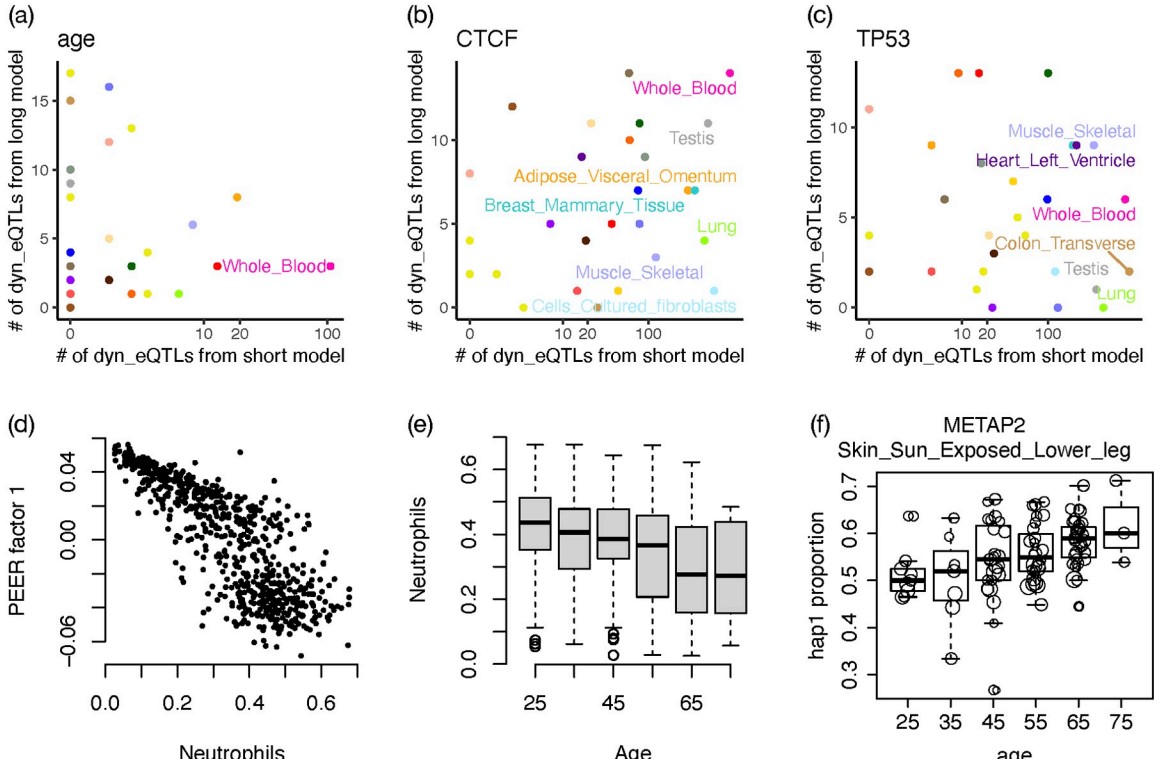

**Fig 4. Dynamic eQTLs.** (a)-(c) The number of dynamic eQTLs identified (q-value < 0.1) using short model (without any additional covariate) versus long model (with 7 additional covariates, top 5 PEER factors and top 2 genotype PCs). X-axis is in log10 scale. Each point is a tissue, and the color scheme is illustrated at the bottom of Fig 2. Tissues with a large number of dynamic eQTLs (> 100 for age or CTCF and > 200 for TP53) using short model are labeled. (d) Association between the first PEER factor from GTEx study and the proportions of neutrophil in whole blood. (e) Association between neutrophil proportion in whole blood and age. (f) An example of dynamic eQTL (q<0.1 in long model) whose eQTL effect size varies with respect to age.

effect size and the variable of interest [18]. As an example, the eQTL effect size on *METAP2* increases with age (Fig 4f) in the long model that accounts for top PEER factors and genotype PCs. Increased expression of *METAP2* is associated with various forms of cancer and it has been investigated as a cancer drug target over the last two decades [22]. Our results show that the strength of genetic regulation of *METAP2*'s expression increases with age, a factor that should be considered when targeting this gene.

We have also assessed whether the genes with dynamic eQTLs with respect to *CTCF* and *TP53* are more likely to be their target genes, as defined by JASPAR database [23]. The annotation data, which was harmonized by harmonizome [24], is a big matrix of size 21,548 × 114, for 21,548 target genes and 114 transcription factors. There are 2,849 targets for *TP53* and only 35 targets for *CTCF*. Using this annotation, we found significant enrichment of *TP53* targets among our dynamic eGenes. Among the 130 genes whose eQTL strength were associated with *TP53* expression, 22 were *TP53* targets while 14 were expected by chance (p-value of Chi-squared test 0.0497). Since the number of targets for *CTCF* was very small, no significant enrichment was found. We also explored the annotated *CTCF* binding sites (CTCFBSDB 2.0 http://insulatordb.uthsc.edu/ [25] and evaluated the overlap between *CTCF*-associated dynamic eGenes and *CTCF* binding sites. We did not find any significant overlap. We suspect this is because the *CTCF* binding sites are highly unspecific. They cover around 28.9% of the

whole genome. Even if we only consider a region of 200 base pair around the center of each annotated binding site, they cover around 7% of the whole genome and the overlap remains insignificant. These results highlight the challenges to interpret the dynamic eQTL results and we expect that additional data and annotation, such as tissue specific activity of transcription factor protein activities (instead of their gene expression) and tissue-specific annotation of target genes, can improve the accuracy and interpretability of the dynamic eQTL results.

## TReCASE has more robust type I error control than RASQUAL

TReCASE and RASQUAL use similar models for total read count data but handle ASE differently. TReCASE models gene-level ASReC for the two haplotypes by a beta-binomial distribution across individuals. In contrast, RASQUAL models ASReC for each SNP by a beta-binomial distribution. For example, considering a gene with ASE measured on 5 SNPs and 100 samples, TReCASE models the gene-level ASReC across the 100 samples by a beta-binomial distribution. In contrast, RASQUAL models the $5 \times 100$ SNP-level ASReCs by a beta-binomial distribution, which effectively inflates the sample size from 100 to 500, leading to inflated type I error. There are also some other less consequential modeling differences between the two methods. For example, RASQUAL assumes the over-dispersion of TReC and ASE are the same while TReCASE estimates them separately, see S1 Text Section B for more details.

We evaluated TReCASE and RASQUAL for eQTL mapping using Geuvadis data [13], see S1 Text Section A.1 for data processing and filtering. Adopting the terminology of RASQUAL, we refer to the SNPs where ASReC are measured as feature SNPs or fSNPs. Applying both methods on the Geuvadis data, TReCASE has higher power than RASQUAL for the genes with less than 10 fSNPs and their power become similar for genes with larger number of fSNPs (Fig 5a). Next, we permuted the SNP genotype data by applying the same permutation for all the SNPs so that the correlations among the SNPs remain unchanged. All the eQTL findings from this permuted dataset should be false positives. We evaluate type I error by examining the proportion of findings with p-values smaller than 0.05, with respect to the number of fSNPs (Fig 5b). TReCASE controls type I error well regardless of the number of fSNPs. In contrast, RASQUAL's type I error increases linearly with the number of fSNPs.

Since there are some other differences between TReCASE and RASQUAL (e.g., RASQUAL handles genotyping error and phasing errors), to confirm the inflated type I error of RASQUAL is mainly due to the fSNP-level beta-binomial distribution assumption, we have implemented a model TReCASE-RL that modifies TReCASE using two assumptions by RASQUAL: fSNP-level beta-binomial distribution and that the over-dispersion of TReC and ASE are the same. We have compared TReCASE, TReCASE-RL and RASQUAL in extensive simulations.

We first considered a situation when SNP-level ASReCs follows a beta-binomial distribution with smaller over-dispersion within a sample and larger over-dispersion across samples. This is a setting where both TReCASE and RASQUAL models are mis-specified since TReCASE assumes within sample over-dispersion is zero while RASQUAL assumes within sample over-dispersion is the same as between sample over-dispersion. In this setting, TReCASE still controls type I error while TReCASE-RL has inflated type I error (Fig 5c).

Our exploration in real data show that in most cases, within sample over-dispersion of ASReCs are zero (S1 Text Section C.5), and thus we focus on this setting in further simulations. We simulated data where the over-dispersion of TReC and ASE were the same so that we could isolate the effect of fSNP-level beta-binomial assumption. Consistent with the findings from Geuvadis data analysis, TReCASE-RL has inflated type I error. This simulation also demonstrates the degree of inflation increases with respect to the number of fSNPs and the size of

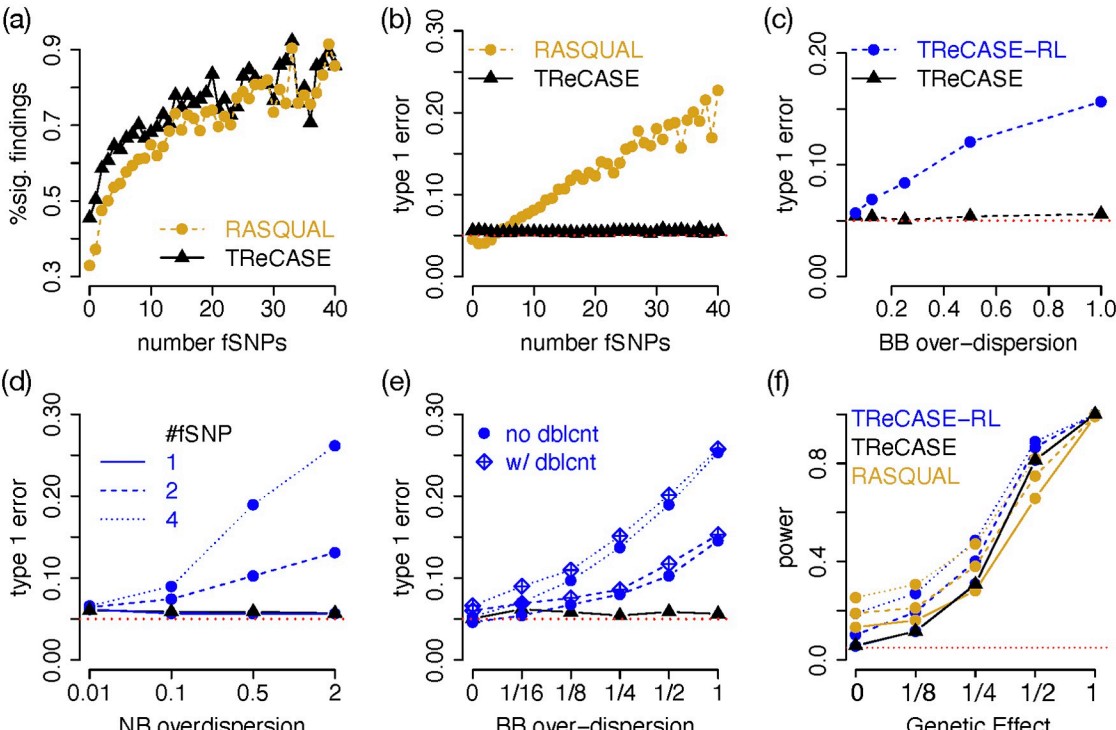

**Fig 5. Compare TReCASE vs. RASQUAL.** (a) Compare the number of significant findings (q-value < 0.05) between TReCASE and RASQUAL for different number of feature SNPs (fSNPs) using Geuvadis data with sample size of 280. (b) The number of significant findings (p-value <0.05) after permuting SNP genotypes, which provides an empirical estimate of type I error. The results of panels (c)-(f) are from simulations with 10,000 replicates. (c) Evaluation of type I error for TReCASE and TReCASE-RL when there is smaller over-dispersion within a sample and larger over-dispersion across samples. We assume there are two heterozygous fSNPs per gene and per sample. Total read counts were simulated with negative binomial with over-dispersion 0.5. The results in (d)-(f) assume there is no over-dispersion across SNPs within an individual. (d) Type I error when the over-dispersion of negative binomial (NB) and beta-binomial (BB) are the same. (e) Effect of double counting. We assume 15% double counting and simulate the data assuming NB over-dispersion to be 0.5. (f) Power analysis when the over-dispersion of NB and BB are both 0.5.

over-dispersion (Fig 5d). When counting allele-specific reads per SNP, some reads may be counted more than once and thus leads to double-counting, which results into inflated type I error, though in a relatively small magnitude (Fig 5e). Finally, we also conducted a power/type I error analysis to compare TReCASE, TReCASE-RL and RASQUAL (Fig 5f). RASQUAL has higher type I error than TReCASE-RL, suggesting some other features of RASQUAL also contribute to type I error inflation. More details of our simulation studies are presented in Section C.6 of S1 Text.

## Discussion

We have demonstrated that eQTL mapping using ASE can substantially improve the power of eQTL mapping than linear regression methods that ignore ASE. When sample size is below 200, the power gain can reach 100%. Even when sample size is as large as 700, using ASE can still improve the power by around 30%. The price to pay for such power gain is extra computational cost. Using 64 threads, one round of eQTL mapping using TReCASE [5] with sample size of 280 is doable within one day. Since computational cost of eQTL mapping using ASE increases roughly linearly with sample size (Fig A25 in S1 Text) and power gain decreases with

sample size and plateaued around 30% when sample size is large than 500 (Fig 2d), the benefit of using ASE for eQTL mapping is easier to justify for studies with smaller sample sizes. In fact, most important findings on the functional roles of eQTLs (e.g., their overlap with GWAS findings) can be accurately quantified using the eQTLs found by a linear model, as demonstrated by earlier GTEx studies [1, 26]. Therefore, one possible choice for eQTL mapping is to apply linear model for the first pass and use the ASE information to validate or refine the eQTL mapping for a subset of genes that warrant further studies. We also want to emphasize although we have re-mapped local eQTLs in 28 GTEx tissues, our results only overlap with a subset of the comprehensive GTEx results that include additional results on distant eQTLs, splice QTLs, cell type-specific eQTLs, and genetic basis of complex diseases etc. [1, 26]

Our geoP method makes it computationally feasible to estimate permutation p-values of TReCASE. Although we have compared geoP with eigenMT [12], it is worth noting that the two methods have very different goals. EigenMT aims to avoid permutations at all and it is computationally very efficient. In contrast, geoP maps eQTLs in permuted data using linear model and uses the results to estimate the permutation p-values for TReCASE. GeoP is computationally faster than doing permutations by TReCASE but it is computationally much more demanding than eigenMT.

We have explored the potential to use ASE to detect dynamic eQTLs. We have found that many dynamic eQTLs identified by a short model that only includes the variable of interest may be confounded by some latent factors such as cell type proportions, which is consistent with the findings from earlier works [19, 20]. There are cases where the meaning of the latent factors is not clear, though they can be captured by the PEER factors. It is an interesting direction for future studies to understand the source of such latent factors.

Another popular method for eQTL mapping using ASE is RASQUAL [7]. We have shown that RASQUAL has inflated type I error. In addition, it is computationally much more demanding than TReCASE. For a dataset with sample size 280 and imputed genotype, it is 10 times slower than TReCASE. For datasets where genotypes are measured by whole genome sequencing (e.g., GTEx data), there is a larger number of heterozygous SNPs where ASE can be measured (Fig A26 in S1 Text), and since RASQUAL handles each SNP separately (while TReCASE works on haplotype level data), it can be 100 times slower than TReCASE. However, RASQUAL has some elegant features (e.g., account for possible sequencing/mapping errors or reference bias). Incorporating these features with the statistical model of TReCASE is a possible direction for a new generation of software package.

In a recent work, Liang et al. [27] proposed a method called mixQTL to combine total expression and ASE for eQTL mapping using a linear model framework. To improve computational efficiency, it uses a linear model framework and assumes the log ratio of allele-specific counts from the two haplotypes follows a normal distribution. This assumption is likely more accurate for genes with larger counts. For example, in their comparison versus standard eQTL mapping method using GTEx whole blood data, they considered 5,734 genes for which (1) at least 15 samples having at least 50 allele-specific counts for each haplotype; and (2) at least 500 samples having a total read count of at least 100. In contrast, we used 16,290 genes with at least 5 samples having at least 5 allele-specific counts. Therefore, count based models like TReCASE can have higher power than mixQTL since they can more effectively exploit ASE in more genes. As a trade-off between power gain and computational time, we agree with Liang et al. [27]'s conclusion that count models are preferred when sample size is relatively small, where higher power gain can over-weight the extra computational time.

## Methods

### Estimation of permutation p-values

When performing local eQTL mapping per gene, we need to scan a large number of SNPs around each gene. The genotypes of these SNPs are often correlated due to linkage disequilibrium. To account for multiple testing across these local SNPs, we can estimate the permutation p-value of the most significant association. It is computationally infeasible to run TReCASE or RASQUAL on a larger number of permuted datasets. Instead, we seek to estimate a relation between permutation p-value and minimum p-value for each gene separately, while using linear regression for eQTL mapping. This is closely related with the concept of "effective number of independent tests" since the ratio between the permutation p-value and corresponding nominal p-value can be considered as the "effective number of independent tests" [9]. Our model shows that the effective number of independent tests of a gene is not a constant. It varies with respect to p-value cutoff.

Let $p_{min,i}$ and $p_{perm,i}$ be the minimum p-value for the $i$-th gene and the corresponding permutation p-value, respectively. [9] observed that there is an approximate linear relation on log scale:

$$E[\log_{10}(p_{perm,i})] = \beta_0 + \beta_1 \, \log_{10}(p_{min,i}). \tag{1}$$

We found such a linear model is accurate when the permutation p-value is small. However, when there are relatively larger permutation p-values, e.g., 0.1, a logistic regression has a better fit:

$$\text{logit}[E(p_{perm,i})] = \beta_0 + \beta_1 \, \log_{10}(p_{min,i}). \tag{2}$$

We implemented a function estimate permutation p-values by automatically produce multiple pairs of minimum p-value and permutation p-value per gene to estimate $\beta_0$ and $\beta_1$ in the logistic regression. Here are more details of the procedure.

1. For each gene we create $k$ new datasets using bootstrap with eQTL effect sizes modified to produce minimum p-values corresponding to permutation p-values in the range from 0.001 to 0.25. In order to approximately achieve a target permutation p-value $\alpha$, we modify the eQTL effect size so that the minimum p-value is $\alpha/E$, where $E$ is a preliminary estimate of the effective number of tests by eigenMT tool [12]. The default value of $k$ is 100. Then the eQTL effect sizes of these 100 datasets are 100 grid points evenly spaced on log scale. We also consider $k = 25, 50,$ and $200$ in our evaluations and conclude that $k = 100$ is a good balance between accuracy and computational efficiency.

2. Run 100 permutations. If more than 40% of the permutation p-values of the bootstrapped data are below the target 0.001, it means some of the eQTL effect sizes in this bootstrap are too large, and we replace them with smaller effect sizes. Alternatively, if more than 30% of the permutation p-values are above 0.3, it means some of the eQTL effect sizes in this bootstrap are too small, and we replace them with bigger effect sizes. We repeat this procedure until most of the p-values are within the range of 0.001 and 0.25.

3. Using the grid selected in the previous step, we run 1,000 permutations for each bootstrapped dataset and calculate permutation p-value of the minimum p-value for each dataset.

Finally, we select the data-points with observed permutation p-value in the range 0 to 0.25 and then fit a linear model (lm) or a generalized linear model (glm, logistic regression) for the

relation between nominal p-value and the corresponding permutation p-value. Using the model fit, we can estimate permutation p-value for any nominal p-value.

## Exploring dynamic eQTLs using individual-specific eQTL effect sizes

We first describe how to define the individual-specific genetic effects using allele-specific expression (ASE) of a gene together with an eQTL. We considered all the genes with permutation p-values smaller than 0.01 and chose the strongest eQTL for each gene. Using ASE, the genetic effect of an eQTL is defined as the proportion of gene expression from the haplotype associated with one allele of the eQTL, which we arbitrarily defined as haplotype 1. Apparently, such genetic effect can only be defined if the eQTL is heterozygous. For the $i$-th individual, denote the random variables for the two ASReCs of haplotype 1 and 2 by $N_{i1}$ and $N_{i2}$, respectively, so that total ASReC for the $i$-th sample is $N_i = N_{i1} + N_{i2}$. Given $N_i$, $N_{i1}$ can be modeled by a beta-binomial distribution as shown in Eq (3).

$$f_{BB}(N_{i1} = n_{i1}; N_i = n_i, \alpha_i, \beta_i) = \binom{n_i}{n_{i1}} \frac{\Gamma(n_{i1} + \alpha_i)\Gamma(n_i - n_{i1} + \beta_i)}{\Gamma(n_i + \alpha_i + \beta_i)} \frac{\Gamma(\alpha_i + \beta_i)}{\Gamma(\alpha_i)\Gamma(\beta_i)}, \tag{3}$$

where $\alpha_i$ and $\beta_i$ are sample specific parameters and they are connected with expected proportion of reads of haplotype 1 (denoted by $\pi_i$) and over-dispersion (denoted by $\theta$) of this beta-binomial distribution by Eq (4):

$$\pi_i = \frac{\alpha_i}{\alpha_i + \beta_i} \ \text{ and } \ \theta = \frac{1}{\alpha_i + \beta_i}. \tag{4}$$

We consider three models, long, medium, and short for different number of covariates included. The long model is:

$$\log\left(\frac{\pi_i}{1 - \pi_i}\right) = \beta_0 + \sum_{u=1}^{2} \beta_{gPCu} \times gPC_{iu} + \sum_{v=1}^{5} \beta_{PFv} \times PF_{iv} + \beta_{cnd} \times cnd_i, \tag{5}$$

where intercept captures average eQTL effect, the seven covariates (two genotype principal components and first 5 PEER factors estimated by GTEx project) capture the interactions between eQTL effect and potential confounders, and $cnd_i$ is the variable of interest. As an illustration, we considered three such variables: age, TP53 and CTCF expression. The set potential confounders is much smaller than set included when considering total read counts (TReC) because the effects of most covariates should be cancelled when comparing the gene expression of one allele versus the other allele. The medium and short models take a subset of the covariates. The medium model includes genotype PCs but not PEER factors. The short model does not include any covariate.

To ensure there is enough data to obtain reliable estimates, we only consider the genes that have enough ASReC ($N_i \geq 10$) in at least 15 individuals. We have explored different implementations of beta-binomial regression and found *R/vglm* provides more numerical stability, especially for the genes with low over-dispersion. Occasionally, when vglm finish with warning, we check whether the likelihood of a beta-binomial model fit is close enough to the likelihood using a binomial fit (within 0.01) and if so, we use the binomial likelihood to refit the model.

## Supporting information

**S1 Table. Enrichment of functional elements among eQTLs identified by MatrixEQTL.**
Each cell in the table shows whether the estimate was significant (at alpha = 0.05) and log fold enrichment.
(CSV)

**S2 Table. Enrichment of functional elements among eQTLs identified by TReCASE.** Format is similar to S1 Table.
(CSV)

**S3 Table. Summary of the dynamic eQTL results with respect to age.** Each row of this table corresponds to a tissue. There are three set of columns: L—model including first two genotype PCs and first 5 PEER factors in addition to the factor of interest, M—model including first two genotype PCs in addition to the factor of interest, and S—model including only the factor of interest. For each model we report four columns: number of significant findings at q-value levels 0.05, 0.10 and 0.25 as well as total number of genes tested.
(CSV)

**S4 Table. Summary of the dynamic eQTL results with respect to CTCF expression.** Format is similar to S3 Table.
(CSV)

**S5 Table. Summary of the dynamic eQTL results with respect to TP53 expression.** Format is similar to S3 Table.
(CSV)

**S6 Table. Supplementary Data for Figs 1–5.** Underlying numerical data for Figs 1–5.
(XLSX)

**S1 Text. Supplementary Methods and Results.** Supplementary Materials for methods and results, including Figs A1-A36, and Tables A1-A18.
(PDF)

## Author Contributions

**Conceptualization:** Vasyl Zhabotynsky, Yi-Juan Hu, Fernando Pardo-Manuel de Villena, Fei Zou, Wei Sun.

**Data curation:** Vasyl Zhabotynsky, Licai Huang, Wei Sun.

**Formal analysis:** Vasyl Zhabotynsky, Licai Huang, Paul Little, Wei Sun.

**Funding acquisition:** Wei Sun.

**Methodology:** Vasyl Zhabotynsky, Paul Little, Yi-Juan Hu, Wei Sun.

**Project administration:** Wei Sun.

**Software:** Vasyl Zhabotynsky, Licai Huang, Paul Little, Wei Sun.

**Supervision:** Wei Sun.

**Writing – original draft:** Vasyl Zhabotynsky, Wei Sun.

**Writing – review & editing:** Vasyl Zhabotynsky, Yi-Juan Hu, Fernando Pardo-Manuel de Villena, Fei Zou, Wei Sun.

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
