## [Decision Letter · Decision Letter 0]

30 Sep 2021

Dear Dr. Sun,

Thank you very much for submitting your Research Article entitled 'eQTL mapping using allele-specific gene expression' to PLOS Genetics.

The manuscript was fully evaluated at the editorial level and by independent peer reviewers. The reviewers appreciated the attention to an important problem, but raised some substantial concerns about the current manuscript. Based on the reviews, we will not be able to accept this version of the manuscript, but we would be willing to review a much-revised version. We cannot, of course, promise publication at that time.

If you decide to revise the manuscript for further consideration at PLOS Genetics, please aim to resubmit within the next 60 days, unless it will take extra time to address the concerns of the reviewers, in which case we would appreciate an expected resubmission date by email to plosgenetics@plos.org.

[LINK]

We are sorry that we cannot be more positive about your manuscript at this stage. Please do not hesitate to contact us if you have any concerns or questions.

Yours sincerely,

Mingyao Li

Associate Editor

PLOS Genetics

David Balding

Section Editor: Methods

PLOS Genetics

Reviewer's Responses to Questions

**Comments to the Authors:**

Reviewer #1: Zhabotynsky et al. present geoP, a new method to approximate permutation p values, to reduce the computational burden of permutation tests in detecting eQTLs with allele-specific gene expression (ASE). The geoP method approximates the permutation p values by estimating the relationship between permutation p values and minimum nominal p values per gene. Overall, the manuscript is well written, and this work has implications for enabling eQTL detection with ASE in large-scale eQTL studies. The authors have devoted efforts in present analyses on 28 GTEx tissues and Geuvadis data. The authors also developed a data processing pipeline to process the RNAseq data saved on cloud. Clarifications and changes that could improve the paper include:

Major:

1. The authors compared the number of eGenes identified by different methods in the real datasets. As the same eGene could be affected by more than one functional regulatory variant, I was wondering for the same gene-SNP pairs tested, how many significant pairs are consistent between different methods, and how many pairs are additionally identified by TReCASE equipped with geoP? How would the results change with different permutation p-value cutoffs?

2. The authors showed the functional categories of the top eQTLs identified by TReCASE and MatrixEQTL are consistent with each other. It would be more convincing to provide additional biological insights by TReCASE beyond the consistent findings.

3. In dynamic eQTL detection, what is the biological interpretation for the identified dynamic eQTLs? For example, how would the effect size of eQTLs change over different expression levels of CTCF or TP53? What functional categories are these dynamic eQTLs enriched in? Are these dynamic eQTLs also detected as eQTLs in the original GTEx study or TReCASE?

4. A recently published paper (Liang et al, 2021, NC) developed a method named mixQTL that can also identify eQTLs with allele-specific expression and is scalable to large sample sizes. What are the benefits of TReCASE equipped with geoP over mixQTL?

5. It would be easier for other people to use your method if it is implemented in software packages. Also, the current description of this method on page 16 is not clear enough. For example, the authors mentioned “do additional adjustment and restart the process”, what kind of “additional adjustment” is suggested?

Minor:

1. In Figure 3e-f, the authors compared functional category enrichment of eQTLs in enhancers in 5 tissues but compared all 28 tissues in figure 3a-d. Why only concentrate on these 5 tissues in figure 3e-f?

2. In figure 5f, why is the detection power increased with decreased genetic effect? Figure 5f is not mentioned in the main text.

Reviewer #2: The authors proposed a geoP method which they claimed to be computationally more efficient and robust at various cutoffs than the eigenMT does in eQTL mapping using ASE information. They also demonstrated the usage of ASE to study dynamic eQTLs and compared two popular methods TReCASE versus RASQUAL. Though the authors did a lot of analyses with various GTEx data sets, I feel the organization and presentation need to be further improved to demonstrate the major contributions of the work. What’s the major contribution and the novelty of the work? They need to be clearly presented and illustrated. I understand the authors try to demonstrate the use of ASE in eQTL mapping. However, the two topics covered in the work, developing a geoP method and studying dynamic eQTLs, do not seem providing a strong support of the title of the work (in fact, they seem to be unrelated). The title is also too broad and not specific.

The abstract needs to be polished to summary the major contribution of the work.

It is not convincible by the way the authors defined false negatives and false positives shown in Figure 1(c). This is also related to what the authors claimed: “Using a permutation p-value cutoff of 0.01 (corresponding to FDR around 1%), we detected 20-100% more eGenes (genes with at least one significant eQTL) than the most recent GTEx study [1], where ASE was not used in eQTL mapping.” Is this claim based on real data analysis or simulation? If based on real data analysis, then it is not convincible. A method that detects more eQTLs than others does not imply that it is more powerful based on the real data analysis. How do you know what you detected are not false positives? geoP does look like have higher false positives than eigenMT does from Fig 1(c) (again, not sure if the figure is based on simulation or real data analysis). To claim a method is more powerful than others, one should do a simulation study in which the underlying truth is known.

The authors stated in the Discussion “Although similar message has been shown in earlier studies, our results are more comprehensive due to the larger number of datasets that we have studied.” Analyzing more datasets does not add to the novelty of the work.

The authors spent large efforts in building an online cloud-based data processing pipeline. This is much appreciated.

Reviewer #3: Review of Zhabotynsky et al.

In this study, Zhabotynsky et al describe and test a method for eQTL mapping that leverages allele-specific read counts to increase mapping power. This is not a novel approach, but the authors do a good job showing that their method is faster and more accurate than other approaches. Overall, I found this paper to be very clearly written with well-justified analyses. This is a very impressive demonstration of the power of utilizing ASE and well-designed statistical modeling to identify eQTL.

Detailed Comments:

1. The authors compare their methods to those reported in the GTEx paper. They note some differences in processing and testing pipeline and write “Despite these minor differences, the number of eGenes reported by the two pipelines (when we use MatrixEQTL for eQTL mapping) are highly consistent (Figure 2a).”. I don’t think Figure 2a shows this. I think Figure 2a needs to have GTEx on one axis and MatrixEQTL on the other in order to support the point they are making. Or is this an axis labelling error and the figure does show what they say? In addition to this figure it would be good to report (numerically) the percentage overlap in eQTL reported by the GTEx paper compared to those discovered using these authors’ pipeline with MatrixEQTL. I assume the overlap is >90-95%.

2. Figure 2a-c what are dotted red lines? It looks like the line in (a) is a best fit regression line, and that this same line is plotted in (b)? The line in (c) looks different but I think that is just due to different axis limits. I suggest harmonizing axis limits on both axes for subfigures (a) through (c)

3. In the section “Explore dynamic eQTLs using individual-specific genetic effects estimated by ASE” (note: typo in section title, should be “Exploring”), I’m not sure I understand the point of including the models that condition on CTCF and TP53. The authors report some interesting results on age-associated dynamic eQTL in blood, but don’t report anything of note regarding the CTCF and TP53 models in the main text.

4. I think it would be very useful if the authors could include some more detailed information on eQTL that they are able to detect that were missing from the MatrixEQTL analysis. For example, they could present a detailed portrait of a single eGene where they show the actual data and illustrate why the ASE and beta-binomial modeling are pulling this eGene to significance.

5. To improve accessibility/usability, I suggest that the authors upload their singularity image to a hub/database such as https://cloud.sylabs.io/. Although there pipeline appears to be available at https://github.com/Sun-lab/gtex_AnVIL, there are instructions for other aspects of analysis included (e.g. cloud analysis) that some users may not use. It would be useful to develop a short example walkthrough for the software.

**Have all data underlying the figures and results presented in the manuscript been provided?**

Reviewer #1: Yes

Reviewer #2: None

Reviewer #3: Yes

PLOS authors have the option to publish the peer review history of their article (what does this mean?). If published, this will include your full peer review and any attached files.

Reviewer #1: No

Reviewer #2: No

Reviewer #3: No

---

## [Decision Letter · Decision Letter 1]

29 Dec 2021

Dear Dr Sun,

Thank you very much for submitting your Research Article entitled 'eQTL mapping using allele-specific count data is computationally feasible, powerful, and provides individual-specific estimates of genetic effects' to PLOS Genetics.

The manuscript was fully evaluated at the editorial level and by independent peer reviewers. The reviewers identified some concerns that we ask you address in a revised manuscript.  Please modify the manuscript according to the review recommendations, address the specific points made by each reviewer.

[LINK]

Yours sincerely,

Mingyao Li

Associate Editor

PLOS Genetics

David Balding

Section Editor: Methods

PLOS Genetics

Reviewer's Responses to Questions

Reviewer #1: I am satisfied with most of the review comments that the authors have addressed, except the results of CTCF and TP53 related dynamic eQTLs. In the updated manuscript, the authors "assessed whether the dynamic eQTLs related with CTCF are enriched among CTCF binding sites (CTCFBSDB 2.0 http://insulatordb.uthsc.edu/ [23]) and whether p53’s target genes (Supplementary Table 3 of [24]) are enriched among the eGenes with dynamic eQTLs with respect to TP53", however, "No significant enrichment was found in either case". The authors chose to study CTCF and TP53 based on the assumption that "TF expression can modulate the strength of eQTLs located in TF binding sites", but the current results do not support this assumption. For the CTCF dynamic eQTLs located in the CTCF binding sites, are their effect sizes more correlated with CTCF expression than other eQTLs? If not, can you find more evidence to support that your identified dynamic eQTLs are truly meaningful?

Reviewer #2: The authors have addressed my comments and I have not further comment.

Reviewer #3: The authors have largely addressed my concerns.

I have two minor suggestions:

(1) I like that they show in Supplemental section C.3 the different examples of eQTL that are missed by MatrixEQTL but captured by TReCASE. But please clarify what part (b) is showing in Supp Figures 11, 12, 13, and 14.

(2) Pipeline usability. The availability of a docker image and the authors’ newer in-depth walkthrough/README are very useful. I would request one small addition. The README at https://github.com/Sun-lab/gtex_AnVIL/blob/master/README.md provides information on the four arguments to their script get_TReC_ASReC.R. It would be really useful to have “toy” versions of those arguments that are files added directly to the GitHub repo. Sometimes when one is running a new pipeline it is very useful to test it on small example files just to ensure that the pipeline is working as expected.

**Have all data underlying the figures and results presented in the manuscript been provided?**

Reviewer #1: Yes

Reviewer #2: Yes

Reviewer #3: Yes

PLOS authors have the option to publish the peer review history of their article (what does this mean?). If published, this will include your full peer review and any attached files.

Reviewer #1: No

Reviewer #2: No

Reviewer #3: No

---

## [Decision Letter · Decision Letter 2]

3 Feb 2022

Dear Dr Sun,

We are pleased to inform you that your manuscript entitled "eQTL mapping using allele-specific count data is computationally feasible, powerful, and provides individual-specific estimates of genetic effects" has been editorially accepted for publication in PLOS Genetics. Congratulations!

Yours sincerely,

Mingyao Li

Associate Editor

PLOS Genetics

David Balding

Section Editor: Methods

PLOS Genetics

Comments from the reviewers (if applicable):

Reviewer #1: The authors have addressed my comments and I have not further comment.

Reviewer #3: The authors have satisfied my concerns - very nice paper!

**Have all data underlying the figures and results presented in the manuscript been provided?**

Reviewer #1: Yes

Reviewer #3: Yes

PLOS authors have the option to publish the peer review history of their article (what does this mean?). If published, this will include your full peer review and any attached files.

Reviewer #1: No

Reviewer #3: No

**Data Deposition**

http://datadryad.org/submit?journalID=pgenetics&manu=PGENETICS-D-21-01129R2

**Press Queries**

---

## [Editor Report · Acceptance letter]

9 Mar 2022

PGENETICS-D-21-01129R2 

eQTL mapping using allele-specific count data is computationally feasible, powerful, and provides individual-specific estimates of genetic effects 

Dear Dr Sun, 

We are pleased to inform you that your manuscript entitled "eQTL mapping using allele-specific count data is computationally feasible, powerful, and provides individual-specific estimates of genetic effects" has been formally accepted for publication in PLOS Genetics! Your manuscript is now with our production department and you will be notified of the publication date in due course.

With kind regards,

Agnes Pap

PLOS Genetics

On behalf of:
